# Increased Alopecia Areata Risk in Children with Attention-Deficit/Hyperactivity Disorder and the Impact of Methylphenidate Use: A Nationwide Population-Based Cohort Study

**DOI:** 10.3390/ijerph18031286

**Published:** 2021-02-01

**Authors:** Hsing-Ying Ho, Chih-Kai Wong, Szu-Yuan Wu, Ray C. Hsiao, Yi-Lung Chen, Cheng-Fang Yen

**Affiliations:** 1Department of Healthcare Administration, College of Medical and Health Science, Asia University, Taichung 41354, Taiwan; hyhftc@gmail.com (H.-Y.H.); szuyuanwu5399@gmail.com (S.-Y.W.); 2Department of Psychology, College of Medical and Health Science, Asia University, Taichung 41354, Taiwan; 3Department of Dermatology, MacKay Memorial Hospital, Taipei 10449, Taiwan; caim1349@gmail.com; 4Department of Food Nutrition and Health Biotechnology, College of Medical and Health Science, Asia University, Taichung 41354, Taiwan; 5Big Data Center, Lo-Hsu Medical Foundation, Lotung Poh-Ai Hospital, Yilan 26546, Taiwan; 6Division of Radiation Oncology, Lo-Hsu Medical Foundation, Lotung Poh-Ai Hospital, Yilan 26546, Taiwan; 7Graduate Institute of Business Administration, Fu Jen Catholic University, Taipei 24205, Taiwan; 8Department of Psychiatry and Behavioral Sciences, University of Washington School of Medicine and Children’s Hospital, Seattle, WA 98105, USA; rhsiao@u.washington.edu; 9Department of Psychiatry, Kaohsiung Medical University Hospital, Kaohsiung 80708, Taiwan; 10Department of Psychiatry, School of Medicine, College of Medicine, Kaohsiung Medical University, Kaohsiung 80708, Taiwan

**Keywords:** alopecia areata, attention-deficit/hyperactivity disorder, methylphenidate

## Abstract

Alopecia areata (AA) is an autoimmune disease that causes sudden hair loss. Although few studies have reported the association between AA and attention-deficit/hyperactivity disorder (ADHD), the impact of methylphenidate (MPH) on AA has not been examined. This study examined whether AA risk is higher in children with ADHD than in those without ADHD as well as the impact of MPH use on AA risk in children with ADHD. From the Taiwan Maternal and Child Health Database, we enrolled all 1,750,456 newborns from 2004 to 2017 in Taiwan. Of them, 90,016 children received a diagnosis of ADHD whereas the remaining 1,660,440 did not. To compare AA risk in ADHD and the impact of MPH treatment on it, multiple Cox regression with adjustments for covariates (i.e., age, sex, and psychiatric comorbidities) was performed. The results indicated that 88 (0.098%) children with ADHD and 1191 (0.072%) children without ADHD had AA. Nevertheless, after adjustment for the covariates, AA risk was higher in children with ADHD than in those without ADHD (adjusted hazard ratio [aHR]: 1.30, 95% confidence interval [CI]: 1.04–1.64). Our data indicated a considerable reduction in AA risk (aHR: 0.64) among children with ADHD who received MPH than among those who did not receive MPH; however, this difference was nonsignificant, indicated by a wide 95% CI (0.32–1.25). In conclusion, ADHD and AA may share some underlying mechanisms.

## 1. Introduction

### 1.1. Alopecia Areata (AA) and Attention-Deficit/Hyperactivity Disorder (ADHD)

Alopecia areata (AA) is an autoimmune condition that attacks the hair follicles, causing nonscarring hair loss [1]. In healthy individuals, hair follicles are immune privilege zones, but in individuals with AA, they can be the target of immune cells such as cytotoxic T cells, natural killer cells, and macrophages [1,2]. AA has a worldwide lifetime incidence of approximately 2% [3]. It can occur at any age, and its incidence increases with age [4]. AA has a substantial disease burden and devastating effects on a patient’s quality of life and self-esteem [5,6].

Attention-deficit/hyperactivity disorder (ADHD) is more likely to be observed in individuals with AA; however, the findings of the association between AA and ADHD are limited and inconsistent [7,8]. For example, a U.S. inpatient survey indicated that compared with patients without AA, patients with AA were more likely to be diagnosed as having ADHD (adjusted odds ratio [aOR]: 8.11) and primarily hospitalized for ADHD (aOR: 16.65) [7]. In contrast, a Danish nationwide study reported that AA was not associated with ADHD [8]. Although the effects were small, the different associations between ADHD and AA across countries may be caused by different cultural contexts [9,10] and the different levels of stress [11]. Furthermore, studies have discussed the association of AA with ADHD from the dermatology perspective, but not from the psychiatric perspective. Additional studies exploring the ADHD–AA association are warranted.

### 1.2. Methylphenidate Use and AA

Methylphenidate (MPH), a stimulant that blocks dopamine and norepinephrine transporters and increases extracellular dopamine and norepinephrine, is used to treat ADHD [12]. Approximately 70% of individuals with ADHD who use methylphenidate present with improvements in their ADHD symptoms [13]. Children with ADHD who use stimulant medications generally have better relationships with peers and family members, perform better in school, are less distractible and impulsive, and have longer attention spans [14].

A case series study recruiting three ADHD cases found that stimulant treatment in individuals with ADHD caused alopecia: all cases demonstrated sudden hair loss (patchy and diffuse hair loss) during the dose-increasing period of MPH (20–30 mg per day), and the hair loss stopped and hair resumed growing after the children were switched from MPH to atomoxetine or lisdexamfetamine [15]. Animal studies have suggested that acute and chronic MPH use could induce oxidative stress, inflammation, and apoptosis in rat brain cells, particularly in the hippocampus [16,17]. Chronic MPH use also increased microglial activation in multiple brain regions in rats, indicating that MPH might trigger inflammatory processes [18]. Oades et al., however, reported that MPH could improve the minor immunological imbalance system in individuals with ADHD [19]. Moreover, low MPH doses can maintain mitochondrial homeostasis in response to hypoxia-induced oxidative stress in neuronal cells [20]. Further research on the MPH use–AA association in individuals with ADHD may aid in elucidating ADHD etiology.

### 1.3. Aims of This Study

To address the aforementioned research gaps, this nationwide population-based cohort study examined whether AA risk was higher in children with ADHD than in those without ADHD and whether MPH use in children with ADHD affected this risk. We hypothesized that AA risk was higher in children with ADHD than in those without ADHD and that AA risk was lower in children with ADHD who received MPH than in those who did not receive MPH.

## 2. Methods

### 2.1. Population

From the Taiwan Maternal and Child Health Database, we extracted the data of all liveborn children from 1 January 2004 to 31 December 2017 including complete information on gestational age at birth and parents’ identities. We recruited children aged >4 years because AA and ADHD are more common and recognizable from that age. The aforementioned database includes data of 99.78% of all births nationwide since 2004 in Taiwan [21]. This study was approved by the Research Ethics Committee of the China Medical University and Hospital (approval number: CMUH108-REC1-142).

### 2.2. Measures

#### 2.2.1. Exposure

The exposure in this study was ADHD diagnosis in children based on the ambulatory care expenditures by visits and the inpatient expenditures by admissions. Participants were considered to have ADHD if they received at least one inpatient diagnosis or more than two outpatient diagnoses of ADHD (International Classification of Disease [ICD], Ninth Revision [ICD-9] code: 314; ICD, Tenth Revision [ICD-10] code: F90) during the study period.

#### 2.2.2. Methylphenidate (MPH) Treatment

MPH treatment in children with ADHD was defined according to dispensation records of the Details of Ambulatory Care Orders, which covers all medication dispensations and accompanying prescriptions for outpatients written in Taiwan since July 1996. MPH use was identified on the basis of the Anatomical Therapeutic Chemical Classification code for MPH (N06BA04) from 1 January 2004 to 31 December 2017. Participants were considered to be taking MPH if they were prescribed it for >3 consecutive months (i.e., ≥90 days between the date of their first and latest prescription).

#### 2.2.3. Outcome

The study outcome was AA diagnosis in children based on the ambulatory care expenditures by visits and the inpatient expenditures by admissions. Participants were considered to have AA if they received at least one inpatient diagnosis or more than two outpatient diagnoses of AA (ICD-9 codes: 704.01 and 704.09, ICD-10 codes: L63.0–L63.2, L63.8, and L63.9) during the study period.

#### 2.2.4. Covariate

Here, the covariates were children’s age, sex, and common mental disorders in children, namely autistic spectrum disorder (ASD; ICD-9 code: 299, ICD-10 code: F84), anxiety disorders (ICD-9 code: 300; ICD-10 codes: F41, F42, and F43.1), tic disorder (ICD-9 code: 307.2, ICD-10 code: F95), and major depressive disorder (MDD; ICD-9 code: 296.2, 296.3, 311, and 300.4; ICD-10 codes: F32, F33, and F34).

### 2.3. Statistical Analysis

SAS (version 9.4) (SAS Institute, Cary, NC, USA) was used for data management and analysis. Descriptive statistics are presented as numbers with proportions and means with standard deviations for categorical and continuous variables, respectively. To compare AA risk between children with and without ADHD, multiple Cox regression was conducted with or without adjustment for age, sex, and comorbid disorders (i.e., ASD, anxiety disorder, tic disorder, and MDD). Age was used as the time scale used in the Cox regression. Furthermore, to clarify the impact of MPH on AA in children with ADHD, multiple Cox regression was conducted with or without adjustments for age, sex, and comorbid disorders in children with ADHD receiving or not receiving MPH. The risks without and with adjustments were presented as crude hazard ratio (cHR) and adjusted hazard ratio (aHR), with their 95% confidence intervals (CIs). If the CI included the null value of 1, the difference was considered nonsignificant. The effect size of HR has been reported based on Azuero’s suggestion, wherein small, medium, and large HRs comparing two groups would be approximately 1.3, 1.9, and 2.8 for categorical risk factors (RR > 1) and 0.77, 0.53, and 0.36 for the categorical protective factors (RR < 1), respectively [22].

## 3. Results

In total, 90,016 records of children with ADHD and 1,660,440 records of children without ADHD were retrieved from the Taiwan Maternal and Child Health Database. Table 1 summarizes their demographics, comorbid disorders, AA, and MPH use among the included children. The ADHD population had a considerably higher proportion of boys (77.41%) than girls. Moreover, children with ADHD had more comorbid disorders (10.87% of ASD, 0.49% of anxiety disorder, 4.93% of tic disorder, and 0.46% of MDD) than did those without ADHD. AA was a relatively rare disease with an incidence proportion of 0.1% in both children with ADHD and without ADHD. Over one third of children with ADHD received MPH.

Table 2 presents the difference in the risks of AA between the ADHD and non-ADHD groups in the whole sample and between MPH users and nonusers among children with ADHD. The crude risk of AA was non-significantly higher in children with ADHD than in those without ADHD (cHR: 1.21; 95% CI: 0.98–1.50). After adjustments for potential confounders (age, sex, and comorbid disorders), AA risk became significantly higher in children with ADHD (aHR: 1.30; 95% CI: 1.04–1.64). Nevertheless, children with ADHD receiving MPH had lower AA risk (cHR: 0.59, 95% CI: 0.30–1.16; aHR: 0.64, 95% CI: 0.32–1.25) than did those not receiving MPH; however, this difference was nonsignificant, as indicated by the wide 95% CI.

## 4. Discussion

### 4.1. AA and ADHD

The findings of this national population-based cohort study revealed that children with ADHD had a higher AA risk than those without ADHD. Some shared inflammatory mechanisms and genetic pathogenesis may explain this association. Individuals with AA had significantly higher levels of inflammatory markers and oxidative stress indicators including serum C-reactive protein (2.52-to 2.73-fold greater), 8-hydroxy deoxyguanosine (1.28-fold greater), and high mobility group box 1 protein (14.87-fold greater) than those without AA [23,24]. Similarly, children with ADHD had an increased level of inflammation indicated by increased serum C-reactive protein and interleukin-6 levels [25].

Major histocompatibility complex genes, particularly the polymorphisms of HLA-DR4 and HLA-DRB1, have been reported to be the potential genes of both AA [26,27] and ADHD [28]. Family studies have indicated that shared genetic causes may exist between ADHD and autoimmune diseases. The mothers of children with ADHD had a higher autoimmune disease risk (aOR: 1.7 of rheumatoid arthritis, aOR: 1.2 of hypothyroidism, aOR: 1.6 of type 1 diabetes, and aOR: 1.8 of multiple sclerosis) than those of children without ADHD [29]. Similarly, ADHD risk was elevated when individuals themselves or their first-degree relatives had autoimmune diseases [30], and the association was reported to be stronger in mothers than in fathers [8].

In addition to AA, the comorbid association of ADHD with other autoimmune diseases including psoriasis, ankylosing spondylitis, autoimmune thyroid diseases, and inflammatory bowel disease has been reported [31,32]. Furthermore, female individuals with ADHD are more likely to have autoimmune comorbidity than male individuals with ADHD [32]. These shared inflammatory mechanisms and genetic pathogenesis between ADHD and autoimmune diseases may explain the increased AA risk in ADHD.

### 4.2. MPH Use and AA

Although we found that AA risk in children with ADHD receiving MPH (aHR: 0.64; 95% CI of aHR: 0.32–1.25) was lower than that in all children with ADHD (aHR: 1.30; 95% CI: 1.04–1.64), the effect of MPH treatment on reducing AA risk was non-significant. AA incidence in children (i.e., 0.1%) was extremely low in the present study; although this study used a national dataset with approximately two million children, only 88 children with ADHD had AA. A study involving children with ADHD indicated that MPH normalizes the immunological characteristics of glial function [19]. Moreover, low MPH doses may protect neuronal cells from hypoxia-induced oxidative stress [20]. Given that stressful life events may trigger AA onset or exacerbation [33,34], MPH treatment may enhance the ability of children with ADHD to manage stress induced by ADHD symptoms and dysfunction in daily lives, thus reducing AA risk. Nevertheless, the association between MPH treatment and AA in individuals with ADHD warrants further research.

### 4.3. Limitations

This study had some limitations. First, our sample comprised children who were 4–14 years old; thus, the associations between ADHD, MPH, and AA may not be generalizable to other age groups. Second, AA incidence in children (i.e., 0.1%) was extremely low in the present study. Third, because of the low number, we excluded children with ADHD taking atomoxetine.

## 5. Conclusions

This national population-based cohort study revealed that children with ADHD had a higher AA risk than those without ADHD. The association between ADHD and AA indicates a possible hypothesis of a shared etiology of immune function between them. The difference in the AA risk between children with ADHD receiving and not receiving MPH was non-significant. The role of MPH treatment on AA in children with ADHD warrants further research.

## Figures and Tables

**Table 1 ijerph-18-01286-t001:** Demographic characteristics, comorbid disorders, alopecia areata, and methylphenidate (MPH) use among the included children.

Variable	ADHD Children	Non-ADHD Children
	N = 90,016	N = 1,660,440
Demographics		
Age, mean (SD)	9.9 (2.3)	9.1 (2.6)
Boy, n (%)	69,679 (77.41)	843,296 (50.79)
Comorbid disorders		
ASD, n (%)	9783 (10.87)	8067 (0.49)
Anxiety disorder, n (%)	443 (0.49)	471 (0.03)
Tic disorder, n (%)	4441 (4.93)	6837 (0.41)
MDD, n (%)	412 (0.46)	378 (0.02)
Alopecia areata, n (%)	88 (0.1)	1191 (0.1)
MPH use, n (%)	31,271 (34.7)	-

Notes. ADHD = attention-deficit/hyperactivity disorder, ASD = autism spectrum disorder, MDD = major depressive disorder, MPH = methylphenidate, SD = standard deviation.

**Table 2 ijerph-18-01286-t002:** Alopecia areata (AA) risk in children with and without attention-deficit/hyperactivity disorder (ADHD) and in children with ADHD receiving and not receiving MPH.

Variable	Crude HR (95% CI)	*p*-Value	Adjusted HR (95% CI) ^a^	*p*-Value
All				
ADHD vs. non-ADHD	1.21 (0.98–1.50)	0.076	1.30 (1.04–1.64)	0.021
ADHD				
MPH users vs. non-users	0.59 (0.30–1.16)	0.126	0.64 (0.32–1.25)	0.207

Notes. HR = hazard ratio, AA = alopecia areata, ADHD = attention-deficit/hyperactivity disorder, MPH = methylphenidate. ^a^ Multiple Cox regression was performed with adjustments for age, sex, and comorbid disorders (i.e., autistic spectrum disorder, anxiety disorder, tic disorder, and major depressive disorder).

## Data Availability

Restrictions apply to the availability of these data. Data were obtained from the Health and Welfare Data Science Center and are available with the permission of the Health and Welfare Data Science Center, Taiwan.

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
