# Peer review of "Increased Alopecia Areata Risk in Children with Attention-Deficit/Hyperactivity Disorder and the Impact of Methylphenidate Use: A Nationwide Population-Based Cohort Study"

_ijerph, 2021, doi:10.3390/ijerph18031286_

Round 1

Reviewer 1 Report

The manuscript "Increased Alopecia Areata Risk in Children with Attention-Deficit/Hyperactivity Disorder and the Impact of Methylphenidate Use: A Nationwide Population-Based Cohort Study" has a very interesting conclusion, i.e. that ADHD and Alopecia Areata may share some underlying mechanisms.

The discussion about the association of ADHD with autoimmune diseases and the shared inflammatory mechanisms and genetic pathogenesis between ADHD and autoimmune diseases is very interesting.

Minor comment/question: Please check page 4, line17: if HLA types in ref 22 should be HLA-DRB1*04 and HLA-DRB1*16, instead of HLA-DR4 and HLA-DRB1, as mentioned in the manuscript.

Author Response

1.      Please check page 4, line17: if HLA types in ref 22 should be HLA-DRB1*04 and HLA-DRB1*16, instead of HLA-DR4 and HLA-DRB1, as mentioned in the manuscript.

Reply:

Because reference 22 reported the associations between AA and HLA-DRB1 polymorphisms, reference 23 reported the associations between AA and HLA-DR4/HLA-DR5 genotypes, and reference 24 reported the associations between ADHD and HLA-DRB1/HLA-DR4 polymorphisms, we described the common findings on HLA types (HLA-DR4 and HLA-DRB1) between AA and ADHD.

To be clearer, we have revised the sentence as follows:

Discussion (lines 187-189):

Major histocompatibility complex genes, particularly the polymorphisms of HLA-DR4 and HLA-DRB1, have been reported to be the potential genes of both AA [22, 23] and ADHD [24].

Reviewer 2 Report

Introduction:

Lines 54 to 62: authors discuss about the findings regardin AA and ADHD in different countries. They did not consider to highlight the possibiltiy that due to culture, habits and so on the association between them might vary across countries?

Lines 72 - 73 "Three ADHD cases..." in which sample size? 3 from 10? 3 from 100? Please, be more specific.

Methods:

Is it the standard rule to diagnose a patient with only one inpatient or at least two outpatient diagnoses?

Results:

Regarding Table 1: "The ADHD population had a considerably higher proportion of boys" (77.41% - please present the frequency here), "children with ADHD had more comodbid disorders.." - would be nice to also have in this palce some frequencies displayed.

From Table 1 to the next paragraph please leave a blank space.

Table 2 - along with CI would be also interesting to have effect sizes, p-or s-value or other measurement of comparison between both groups.

Discussion:

Lines 173- 174: "Individuals with AA have hogher levels of inflamatory..." how big are those higher levels? Some reference value would be interesting to present. The same for "The mothers of children with ADHD had a higher autoimmune disease risk than did those of children without ADHD" - Line 182.

Author Response

Introduction:

1.      Lines 54 to 62: authors discuss about the findings regardin AA and ADHD in different countries. They did not consider to highlight the possibiltiy that due to culture, habits and so on the association between them might vary across countries?

Reply:

        We have highlighted the possible influence of cultural contexts in the manuscript. The revised paragraph and references that we cited are presented as follows:

Introduction (lines 54-62):

Attention-deficit/hyperactivity disorder (ADHD) is more likely to be observed in individuals with AA; however, the findings of the association between AA and ADHD are limited and inconsistent [7, 8]. For example, a US inpatient survey indicated that compared with patients without AA, patients with AA were more likely to be diagnosed as having ADHD (adjusted odds ratio [aOR]: 8.11) and primarily hospitalized for ADHD (aOR: 16.65) [7]. By contrast, a Danish nationwide study reported that AA was not associated with ADHD [8]. Although the effects were small, the different associations between ADHD and AA across countries may be caused by different cultural contexts [31, 32] and the different levels of stress [33].

References:

  1. Norvilitis, J. M.; Fang, P. Perceptions of ADHD in China and the United States: A preliminary study. J Atten Disord 2005, 9(2), 413-424. doi: 10.1177/1087054705281123.
  2. Timimi, S.; Taylor, E. ADHD is best understood as a cultural construct. Br J Psychiatry 2004, 184(1), 8-9. doi: 10.1192/bjp.184.1.8.
  3. Hwang, S.; Shin, J.; Kim, T. G.; Kim, D. Y.; Ho Oh, S. Large-scale retrospective cohort study of psychological stress in patients with alopecia areata according to the frequency of intralesional steroid injection. Acta Derm Venereol 2019, 99(2), 236-237. Doi: 10.2340/00015555-3079.

2.      Lines 72 - 73 "Three ADHD cases..." in which sample size? 3 from 10? 3 from 100? Please, be more specific.

Reply:

Reference 12 that we cited is a case series study with 3 of sample size regarding AA and MPH use. To be clearer, we have revised the paragraph as follows:

Introduction (lines 73-77):

A case series study recruiting three ADHD cases found that stimulant treatment in individuals with ADHD causes alopecia: All three ADHD cases demonstrated sudden hair loss (patchy and diffuse hair loss) during the dose-increasing period of MPH (2030 mg per day), and the hair loss stopped and hair resumed growing after the children were switched from MPH to atomoxetine or lisdexamfetamine [12].

Methods:

3.      Is it the standard rule to diagnose a patient with only one inpatient or at least two outpatient diagnoses?

Reply:

        It has been proposed that the use of multiple outpatient records or at least one record for a diagnosis of a disease has an improvement in its accuracy to avoid misclassification (Lin et al., 2005), although no validation study has been examined in ADHD and AA. Another advantage of the requirement of the multiple records of diagnoses is to ensure the duration of symptoms at least last for a certain time, not just met the symptoms criteria. Some studies have also used multiple records of the diagnosis of ADHD to define their participants as having ADHD to increase the diagnosis accuracy (Montejano et al., 2011, Chen et al., 2020).

References:

Lin, C.-C., Lai, M.-S., Syu, C.-Y., Chang, S.-C., & Tseng, F.-Y. (2005). Accuracy of diabetes diagnosis in health insurance claims data in Taiwan. Journal of the Formosan Medical Association, 104(3), 157-163.

Montejano, L., Sasané, R., Hodgkins, P., Russo, L., & Huse, D. (2011). Adult ADHD: prevalence of diagnosis in a US population with employer health insurance. Current Medical Research and Opinion, 27(sup2), 5-11.

Chen, V. C.-H., Yao-Hsu Yang, Ting Yu Kuo, Mong-Liang Lu, Wei-Ting Tseng, Tsai-Yu Hou, . . . Gossop, M. (2020). Methylphenidate and the risk of burn injury among children with attention-deficit/hyperactivity disorder. Epidemiology and Psychiatric Sciences, 29(e146), 1–7.

Results:

4.      Regarding Table 1: "The ADHD population had a considerably higher proportion of boys" (77.41% - please present the frequency here), "children with ADHD had more comodbid disorders.." - would be nice to also have in this palce some frequencies displayed.

Reply:

        We have revised the presentation of Table 1 as follows:

Results (lines 150-153):

The ADHD population had a considerably higher proportion of boys (77.41%) than girls. Moreover, children with ADHD had more comorbid disorders (10.87% of ASD, 0.49% of anxiety disorder, 4.93% of tic disorder, and 0.46% of MDD) than did those without ADHD.

5.      From Table 1 to the next paragraph please leave a blank space.

Reply:

We have edited the arrangement between Table 1 and the text.

6.      Table 2 - along with CI would be also interesting to have effect sizes, p-or s-value or other measurement of comparison between both groups.

Reply:

        We have added P-values into Table 2. Moreover, we have added the effect size of HR into the method section as follows:

Statistical Analysis (lines 142-145):

The effect size of HR has been reported based on Azuero's suggestion, wherein small, medium, and large HRs comparing 2 groups would be approximately 1.3, 1.9, and 2.8 for categorical risk factors (RR > 1) and 0.77, 0.53, and 0.36 for categorical protective factors (RR < 1), respectively [34].

Discussion:

7.      Lines 173- 174: "Individuals with AA have hogher levels of inflamatory..." how big are those higher levels? Some reference value would be interesting to present. The same for "The mothers of children with ADHD had a higher autoimmune disease risk than did those of children without ADHD" - Line 182.

Reply:

        We have added the values of references to present the inflammatory levels and the disease risks more clearly. The revised paragraph is presented as follows:

Discussion (lines 181-185):

Individuals with AA had significantly higher levels of inflammatory markers and oxidative stress indicators, including serum C-reactive protein (2.52-to 2.73-fold greater), 8-hydroxy deoxyguanosine (1.28-fold greater), and high mobility group box 1 protein (14.87-fold greater), than those without AA [19, 20].

Discussion (lines 190-193):

The mothers of children with ADHD had a higher autoimmune disease risk (aOR: 1.7 of rheumatoid arthritis, aOR: 1.2 of hypothyroidism, aOR: 1.6 of type 1 diabetes, and aOR: 1.8 of multiple sclerosis) than did those of children without ADHD [25].

Reviewer 3 Report

Your paper is interesting and well written. I have only one concern. In the conclusions you say that the association between ADHD and AA indicates a shared etiology of immune function. Other studies that you refer to report data that could indicate a shared etiology of immune function, but you have not studied immune functions in your own research project, so I do not think that you can say anything about the etiology from the results of your study. All you can say is that the hypothesis of a shared etiology of immune function is compatible with your results since you found an association between ADHD and AA.

Author Response

1.      Your paper is interesting and well written. I have only one concern. In the conclusions you say that the association between ADHD and AA indicates a shared etiology of immune function. Other studies that you refer to report data that could indicate a shared etiology of immune function, but you have not studied immune functions in your own research project, so I do not think that you can say anything about the etiology from the results of your study. All you can say is that the hypothesis of a shared etiology of immune function is compatible with your results since you found an association between ADHD and AA.

Reply:

        Thanks for the comments, we have revised the paragraph in the Conclusion section accordingly as follows:

Conclusions (lines 223-228):

This national population-based cohort study revealed that children with ADHD had a higher AA risk than did those without ADHD. The association between ADHD and AA indicates a possible hypothesis of a shared etiology of immune function between them. The difference in the AA risk between children with ADHD receiving and not receiving MPH was nonsignificant. The role of MPH treatment on AA in children with ADHD warrants further research.